TOPICAL REVIEW

# The physiology, anatomy and stimulation of the vagus nerve in epilepsy

Mikaela Patros[1], Shobi Sivathamboo[1,2,3,4], Hugh D. Simpson[1,2] , Terence J. O'Brien[1,2,3,4] and Vaughan G. Macefield[1,5] 

[1]*Department of Neuroscience, School of Translational Medicine, Monash University, Melbourne, Victoria, Australia*
[2]*Department of Neurology, The Alfred Hospital, Melbourne, Victoria, Australia*
[3]*Department of Medicine, The Royal Melbourne Hospital, The University of Melbourne, Parkville, Victoria, Australia*
[4]*Department of Neurology, The Royal Melbourne Hospital, Parkville, Victoria, Australia*
[5]*Department of Cardiometabolic Health, The University of Melbourne, Parkville, Victoria, Australia*

Handling Editors: Laura Bennet & Christopher Lear

The peer review history is available in the Supporting Information section of this article (https://doi.org/10.1113/JP287164#support-information-section).

**Abstract figure legend** The vagus nerve is both the longest cranial nerve and the largest, providing parasympathetic control of multiple organ systems within the thorax and abdomen, including the cardiovascular, respiratory and gastrointestinal systems (Ottaviani & Macefield, 2022). Stimulation of the vagus nerve is beneficial in a number of human diseases, most notably epilepsy. Through consideration of the anatomical composition of the vagus nerve, its physiology and its distribution throughout the body, the effects of vagus nerve stimulation (VNS) in the context of drug-resistant epilepsy are considered. The review is divided into two sections: part one surveys the anatomy and physiology of the vagus nerve, and part two considers stimulation of the vagus nerve. AMY, amygdala; DRN, dorsal raphe nucleus; HIP, hippocampus; LC, locus coeruleus; NTS, nucleus tractus solitarius; PB, parabrachial nucleus; PFC, prefrontal cortex.

The Journal of Physiology

**Abstract**  The vagus nerve is the longest cranial nerve, with much of its territory residing outside the head, in the neck, chest and abdomen. Although belonging to the parasympathetic division of the autonomic nervous system, it is dominated by sensory axons originating in the heart, lungs and airways and the gastrointestinal tract. Electrical stimulation of the cervical vagus nerve via surgically implanted cuff electrodes has been used clinically for the treatment of drug-resistant epilepsy for three decades but has also shown efficacy in the treatment of drug-resistant depression and certain gastrointestinal disorders. Through consideration of the anatomical composition of the vagus nerve, its physiology and its distribution throughout the body, we review the effects of vagus nerve stimulation in the context of drug-resistant epilepsy. This narrative review is divided into two sections: part one surveys the anatomy and physiology of the vagus nerve, and part two describes what we know about how vagus nerve stimulation works.

(Received 2 September 2024; accepted after revision 12 February 2025; first published online 8 March 2025)

**Corresponding author** Vaughan G. Macefield: Department of Neuroscience, School of Translational Medicine, Monash University, 99 Commercial Road, Melbourne, VIC 3004, Australia.    Email: vaughan.macefield@monash.edu

## Part one: anatomy and physiology of the vagus nerve

**Autonomic control and composition of the vagus nerve.** Named from the Latin word 'wandering', the vagus nerve exits through the jugular foramen of the skull and descends along the neck with both the carotid artery and the internal jugular vein (Walker, 1990). Within the neck, the vagus nerve gives off branches that innervate the trachea and supply some of the muscles of the pharynx and larynx (Celik Gokyigit et al., 2016). Following on, the vagus then travels down through the thoracic and abdominal cavities to innervate important organ systems, such as the heart, lungs and airways and the gastro-intestinal tract (Câmara & Griessenauer, 2015; Ottaviani & Macefield, 2022). The vagus nerve travels within the carotid sheath along with the carotid artery, which typically lies medial to the nerve, and the internal jugular vein, which is positioned superficial to the nerve (Robbins et al., 2008). The carotid sheath is located anterior to the cervical sympathetic trunk, and there is some dispute over whether the arrangement of these structures remains consistent along the course of the neck; specifically, the assumed dorsolateral position of the vagus nerve between the carotid artery and internal jugular vein (Câmara & Griessenauer, 2015). Cadaveric investigations have shown that the position of the vagus nerve within the carotid sheath is variable (Hammer et al., 2018). Planitzer et al. (2017) performed dissections of 35 cadavers and showed that the position of the cervical vagus often differed from the anticipated dorsolateral arrangement. Moreover, only 42% of the observed anatomical arrangements of the cervical vagus were dorsolateral between the carotid artery and internal jugular vein (Planitzer et al., 2017).

In most species, the vagus nerve, like somatic peripheral nerves, has a fascicular organization, in which individual nerve fibres are bundled together and surrounded by connective tissue called endoneurium. These nerve bundles are compartmentalized by a connective network of perineurium to form individual nerve fascicles (Olshansky et al., 2008). Existing knowledge of the structure of the vagus nerve has been informed in large part by animal investigations, though there is remarkable variation in the number of fascicles across species; the mouse contains one, whereas (as shown in Fig. 1) the pig contains >30 (Ottaviani & Macefield, 2022; Stakenborg et al., 2020). In humans, cadaveric investigations have revealed the mean number of fascicles within the cervical vagus to be between 5 and 10 (Seki et al., 2014; Stakenborg et al., 2020).

The vagus nerve is mixed in its composition, containing both afferent and efferent fibres. Afferent fibres, carrying visceral sensory information, are the dominant fibre type within the vagus nerve (Aalbers et al., 2011). These can be classified into three groups: (i) general somatic

**Mikaela Patros** is a third year PhD candidate in the Department of Neuroscience at Monash University, Melbourne. She completed her Bachelor of Science with First Class Honours in 2021 with The University of Melbourne, performing micro-electrode recordings from the human vagus nerve under the supervision of Professor Vaughan Macefield. Her PhD project, also under the supervision of Vaughan Macefield, is assessing the effects of clinical vagus nerve stimulation on muscle sympathetic nerve activity in patients with drug-resistant epilepsy. She is also performing microelectrode recordings from the cervical vagus nerve in the same patients to identify which axons are activated by the vagus nerve stimulation.

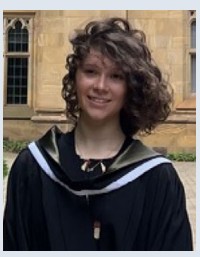

afferents responsible for cutaneous sensation; (ii) general visceral afferents carrying information from cervical structures and thoracic and abdominal viscera; and (iii) special visceral afferents responsible for taste. Importantly, general visceral afferents innervate baroreceptors located in the aortic arch, which convey blood pressure changes to the CNS. General visceral afferents also innervate chemo-receptors in the aortic arch, which detect low blood $O_2$ and high $CO_2$ (Câmara & Griessenauer, 2015). Efferent fibres can be categorized as either general visceral efferents or special visceral efferents (Ottaviani & Macefield, 2022). General visceral efferents innervate the smooth muscle and glands of the thoracic and abdominal viscera, pharynx and larynx, whereas special visceral efferents innervate the striated muscle of the pharynx and larynx. General visceral efferent fibres originate in the nucleus ambiguus (NA), and special visceral efferents arise from neurones in the dorsal motor nucleus of the vagus (DMNV) (Ruffoli et al., 2011).

Based on electrical properties, the vagus nerve is composed of three fibre types (A, B and C). The A and B fibres are myelinated and larger in diameter than C fibres, which are small and unmyelinated (Ottaviani & Macefield, 2022; Stakenborg et al., 2020). The C fibres have a much higher electrical threshold for excitation and have slow conduction velocities ($\sim$1 m/s). Unmyelinated fibres make up $\sim$80% of the total number of fibres within the vagus nerve in both humans and animals (Helmers et al., 2012). These fibres can function as afferent fibres, which transmit sensory information from the viscera to the vagal nuclei of the medulla, or efferent fibres, which carry motor signals from the medulla to effector organs within the thoracic and abdominal cavities. The A and B fibres have electrical thresholds of 0.02–0.2 and 0.04–0.6 mA, respectively. Unmyelinated C fibre activation requires

currents that are expected to be >2 mA (Ottaviani & Macefield, 2022). These values have particular significance when considering clinical vagus nerve stimulation (VNS) in the treatment of epilepsy. Stimulation parameters for VNS will be discussed in greater detail later in this review.

**Vagal afferent network within the brain.** There are four sets of bilateral vagal nuclei within the medulla. Afferent fibres project to the nucleus tractus solitarii (NTS) and the spinal nucleus of the trigeminal nerve, with most visceral afferents synapsing within the NTS, which is the primary receiving area for visceral sensory information. Efferent fibres originate in either the NA or the DMNV (Chae et al., 2003). The NTS projects to many structures located within the posterior fossa, such as the other nuclei of the dorsal medulla, the parabrachial nucleus and pontine nuclei, the vermis and other areas of the inferior cerebellar hemi-spheres (Henry, 2002). Second-order neurones within the parabrachial nucleus then project to a variety of structures, including the hypothalamus, amygdala, anterior insula, infralimbic cortex, thalamus and lateral prefrontal cortex. Through its indirect projection to the amygdala, the NTS can modulate pathways of the limbic system involving the amygdala, hippocampus and entorhinal cortex (Van Bockstaele et al., 1999). Moreover, the NTS projects to the raphe nuclei and locus coeruleus (LC), regions which host serotonergic and noradrenergic cell bodies, respectively (Nemeroff et al., 2006). The adrenergic innervation of the LC and serotonergic innervation of the raphe are widespread, over the entire cortex. Further to this, direct projections also exist between the NTS and the cerebellum and periaqueductal grey (Nemeroff et al., 2006). General somatic afferent fibres terminate in the spinal nucleus of the trigeminal nerve, whereas general visceral afferents and special visceral afferents project to the NTS, which plays a key role in integrating neuronal input to regulate autonomic processes, such as cardiorespiratory function (Loewy & Spyer, 1990).

**Vagal innervation of the lungs and airways.** The lower airways are regulated in order to allow optimal gas exchange in normal physiological conditions (Mazzone & Canning, 2013). Dynamic control of airway tone occurs through innervation of the bronchiolar smooth muscle that lines the airways. Parasympathetic and sympathetic innervation of the airways varies considerably across different species (Mazzone & Canning, 2013). For example, in dogs and cats the sympathetic innervation has an inhibitory influence over the airways (Barnes & Liu, 1995; Cabezas et al., 1971). The smooth muscle lining human and guinea-pig airways, however, is innervated by parasympathetic postganglionic neurones that control the relaxation and opening of the airways; in fact, it has been shown that the smooth muscle lining the airways

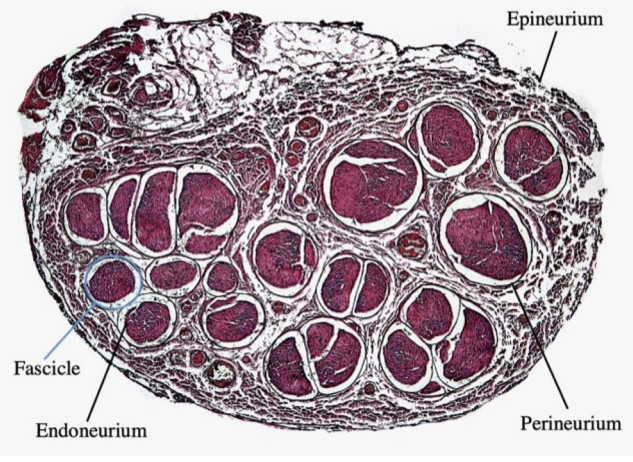

**Figure 1. Transverse histological section of a pig cervical vagus nerve, stained with Haematoxylin and Eosin**
Reproduced with permission from Ottaviani and Macefield (2022).

in humans is devoid of any sympathetic innervation (Canning, 2006; Richardson & Béland, 1976).

Early neurophysiological studies of the vagus nerve revealed activity related to respiration, with pulmonary vagal afferent fibres in which maximal activity was recorded during inspiration and lowest activity during expiration; it was proposed that these receptors had an inhibitory influence on respiratory centres in the brainstem (Adrian, 1933). This evidence suggested that vagal fibres were in receipt of mechanosensory information pertaining to stretch receptors within the lungs and airways (Paintal, 1955; Yu, 2020). There are two main types of pulmonary stretch receptors responsible for different aspects of respiratory function. These are slowly adapting receptors, activated during a sustained period of lung inflation, and rapidly adapting receptors that fire promptly in response to lung inflation and deflation but also to mechanical stimulation (Adrian, 1933; Widdicombe, 2003). Another type are the irritant receptors located in the larynx, which function in a similar manner to rapidly adapting receptors but are also stimulated in response to chemical stimuli (Widdicombe, 2003). Irritant receptors play a key role in the formation of defensive reflex mechanisms, such as cough (Mazzone, 2005). The main difference between these irritant receptors and the slowly adapting and rapidly adapting mechanoreceptors is that the irritant receptors do not fire in response to changes in pressure, stretch or contraction of the airways (Canning, 2010).

**Vagal innervation of the heart and great vessels.** Interestingly, the vagus nerve happens to be the only bilateral human nerve that delivers a distinct asymmetric supply to organs such as the heart (Ottaviani & Macefield, 2022). The vagus nerve extends to provide nodal innervation of the heart, whereby vagal efferents contribute to the modulation of sinoatrial (SA) nodal rate and atrioventricular (AV) conduction time (Furukawa et al., 1991). The SA node is the primary intrinsic pacemaker of the heart, responsible for setting the heart rate. The AV node is responsible for maintaining a delay between atrial and ventricular depolarization to permit sufficient time for the ventricles to fill (Sassa & Miyazaki, 1920). Fibres within the right vagus nerve innervate the SA node, and fibres within the left vagus nerve innervate the AV node (Cheng & Powley, 2000; Cheng et al., 1999). Animal investigations have demonstrated that electrical stimulation of the right cervical vagus nerve had a more profound effect on the chronotropic function of the SA node in comparison to stimulation of the left vagus nerve (Ardell & Randall, 1986). When comparing stimulation of the right and left vagus nerve in the rabbit, it was shown that right VNS was more effective at producing bradycardia than left VNS (Brack et al., 2004). It should be noted, however, that that in some animal models stimulation of the left vagus nerve has been shown to evoke bradycardia. Selective stimulation of the left vagus nerve bundle in rats significantly reduced heart rate by a varying degree, with changes in stimulation frequency, amplitude and pulse width (Gierthmuehlen & Plachta, 2016). In pigs, intraneural closed-loop cervical stimulation of the left vagus nerve resulted in substantial reductions of heart rate (Sevcencu et al., 2018). The electrodes used by Sevcencu et al. (2018), however, were sutured longitudinally through the nerve, most probably leading to selective activation of fibres displaying chronotropic cardiac modulation. It is important to mention that achieving bradycardia with clinical left-sided VNS is highly unlikely, given that stimulation is neither selective nor intraneural and noting that the left vagus nerve innervates the AV node, not the SA node.

Stimulation of the right vagus nerve in cats resulted in a longer P–P interval, producing an effective slowing of the heart. Stimulation of the left vagus nerve, however, produced a lengthening of AV conduction duration (Tsuboi et al., 2000). Stimulation of the vagus nerve has also been shown to promote lusitropy, increasing the rate of relaxation (Henning et al., 1996). Moreover, vagal innervation of the ventricular walls has been shown to modulate ventricular contraction by lengthening the ventricular refractive period and decreasing mean arterial pressure (Martins & Zipes, 1980). Hence, the vagus nerve is thought to contribute to cardiovascular function by having a negative inotropic effect, diminishing the force of ventricular contraction and reducing blood pressure.

Vagal afferents contribute to the control of the cardiovascular system via several reflexes. One such reflex is the arterial baroreflex (Buijs et al., 2013). Mechanoreceptors subserving this reflex are located within the aortic arch and carotid sinus, increasing their firing during radial distension of the vessels during systole. Early work by Adrian (1933) investigated vagal fibres within the cat and described what are now referred to as baroreceptor afferents. These vagal fibres demonstrated an increase in activity, corresponding to rises in blood pressure. Carotid sinus baroreceptor afferents travel within the glossopharyngeal nerve, whereas baroreceptor afferents in the aortic arch are conveyed by the vagus nerve (Bianchi-da-Silva et al., 2000); both synapse within the NTS in the medulla.

Activation of these vagal afferent fibres can reflexly modulate both sympathetic and parasympathetic efferent activity, via the arterial baroreflex. Key to the efficacy of this reflex is the activation of vagal efferent fibres to the SA node of the heart, which induces bradycardia, and inhibition of sympathetic vasoconstrictor neurones. In normal circumstances, the phasic activation of baroreceptor afferents during systole evokes vagal efferent activity to the heart to decrease its rate and contractility, whereas unloading of the baroreceptors has the

opposite effect. Conversely, the arterial baroreceptors inhibit sympathetic outflow to the heart (if present), whereas unloading has the opposite action (Karemaker & Wesseling, 2008). Thus, it is evident that the sympathetic and parasympathetic systems interact in a complex way to maintain normal cardiac function (Olshansky et al., 2008).

In the resting state, it is ongoing parasympathetic activity that dominates over sympathetic activity in relationship to cardiac function. In the setting of cardiac disease, such as heart failure, sympathetic outflow to the heart and to the muscle, splanchnic and renal vascular beds is increased, contributing to the progression of the disease, whereas parasympathetic (vagal) outflow to the heart is believed to be reduced (Dunlap et al., 2003). Parasympathetic withdrawal is prominent in early stages of the disease, even when sympathetic activity continues to increase over time (Kinugawa & Dibner-Dunlap, 1995). It is also well established that patients with heart failure have abnormal baroreceptor control and have reduced baroreflex sensitivity (Dibner-Dunlap & Thames, 1989) and that carotid chemoreceptor activity is elevated in the heart failure state (Sun et al., 1999). It therefore makes sense that targeting the vagus nerve might be beneficial in the management and treatment of cardiovascular dysfunction.

Vagal afferents also innervate chemoreceptors found in the aortic arch, responsible for monitoring changes in blood pH, oxygen and carbon dioxide levels (Wang et al., 2019). Peripheral chemoreceptors respond to hypoxic (low-oxygen) states, which leads to an increase in ventilation. The Bezold–Jarisch reflex is another cardiac reflex that is thought to be mediated by activation of these chemoreceptors and mechanoreceptors located in the ventricular walls (Mark, 1983). Stimulation of the vagal afferent fibres involved in this reflex results in an increase in cardiac parasympathetic activity and a decrease in cardiac sympathetic drive, leading to a slowing of heart rate (bradycardia), decrease in blood pressure (hypotension) and vasodilatation. There is some debate, however, regarding whether ventricular mechanoreceptors are involved in this regulatory reflex. Work in anaesthetized dogs by Drinkhill et al. (2001) has called into question the role of these receptors in maintaining normal cardiac function. Given the absence of reflex vasodilatation in the presence of reduced ventricular filling and increases in force of contraction in their experiments, the authors concluded that ventricular mechanoreceptors are not activated in this way and thus do not contribute to this reflex and, more broadly, to regulating cardiovascular function within normal limits.

The Bainbridge reflex functions in severe diseased states, such as heart failure or myocardial ischaemia (Pakkam & Brown, 2022). It is also possible that this reflex plays a part in the generation of respiratory sinus arrythmia, though there is some debate regarding whether this is likely, given the high atrial pressures used experimentally to produce this reflex (Ottaviani & Macefield, 2022).

When these reflexes fail, regular autonomic processing is impaired, leading to atypical cardiovascular function. The parasympathetic and sympathetic divisions of the autonomic nervous system have divergent influences on autonomic activity, and a balance must be maintained for optimal function. Consequently, any imbalance present within the autonomic divisions of the nervous system should be addressed in order to provide insight on how to explain and possibly treat certain cardiovascular complications (Kobayashi et al., 2013).

**Vagal innervation of the gastrointestinal tract.** Below the diaphragm the two vagus nerves change their orientation: the left now lies anterior to the oesophagus, while the right is located posteriorly (Ottaviani & Macefield, 2022). The majority of axons at this point are unmyelinated C fibres, and ∼90% at the abdominal level are afferent fibres. Broadly speaking, vagal innervation of the gastrointestinal tract regulates digestion, promotes peristalsis, relaxes sphincters and increases glandular secretions of the liver and pancreas (Câmara & Griessenauer, 2015; Chang et al., 2003; Iggo & Leek, 1967). These actions are carried out through vagal innervation of enteric ganglia located in regions such as the pancreas, gallbladder and stomach (Furness et al., 2022).

The C fibres in the gastrointestinal tract have endings specifically in a layer of connective tissue called the lamina propria, in the intestinal and gastric mucosa. These mucosal endings are considered chemoreceptors and are sensitive to pH and to osmotic and chemical stimuli from the lumen (Ottaviani & Macefield, 2022). The vagus also innervates the smooth muscle of the gut, exocrine glands and endocrine bodies of the mid- and upper gut (Bonaz et al., 2017; Câmara & Griessenauer, 2015; Iggo & Leek, 1967; Williams et al., 2016). Afferent vagal endings in the outer layers of muscle and myenteric plexus are classed as mechanoreceptors because they are sensitive to gastric distension (Powley & Phillips, 2011).

Sensory information from vagal afferent chemoreceptor and mechanoreceptor endings is transmitted to the NTS, where these signals project to the DMNV, which is responsible for relaying efferent vagal output to regulate metabolic function (Berthoud & Neuhuber, 2000). Experiments in rats have shown that direct electrical stimulation of the DMNV results in elevated levels of insulin, gastric acid and glucagon secretion (Laughton & Powley, 1987). Likewise, electrical stimulation of the cervical vagus nerve has been shown to increase insulin, gastric acid and glucagon levels in rats (Berthoud & Powley, 1987; Nishi et al., 1987), confirming the vital role of the vagus nerve in supporting gastrointestinal function.

## Part 2: stimulation of the vagus nerve in epilepsy

Electrical stimulation of the vagus nerve has been used as a treatment for a number of neurological and non-neurological human diseases. Stimulation of the vagus extends to psychiatric, gastrointestinal and immunological disease (Bonaz et al., 2017; Koopman et al., 2016; Rush et al., 2000); however, this section of the review will focus on its clinical applications in the context of drug-resistant epilepsy.

**Epilepsy, mortality and drug resistance.** Epilepsy is one of the most common and disabling neurological conditions, affecting >70 million people worldwide (Thijs et al., 2019), and is characterized by an enduring predisposition of the brain to generate recurrent, unprovoked seizures. Seizures involve abnormal or excessive hypersynchronous neuronal activity in the brain associated with transient signs and/or symptoms (Fisher et al., 2005; Moshé et al., 2015). There are many direct and indirect consequences from epilepsy, including decreases in quality of life and neurocognitive functioning and increased psychosocial burden. Furthermore, epilepsy is associated with an increased risk of all-cause mortality, including sudden unexpected death in epilepsy (SUDEP), which is the leading cause of mortality directly from epilepsy (Kløvgaard et al., 2022).

Proposed mechanisms from monitored SUDEP cases suggest that there is an early, centrally mediated respiratory arrest, which then drives terminal cardiac arrest, immediately following a tonic–clonic seizure (Shorvon & Tomson, 2011). Heart rate variability (HRV) is used routinely as an indirect measure of autonomic control over the cardiac cycle (Malliani et al., 1991). Altered HRV has been shown to be a predictive biomarker for sudden death in mostly cardiovascular disease populations, such as heart failure (Kleiger et al., 1987). Aberrant HRV has also been observed in a number of SUDEP cases, with some reports of acute vagal dysfunction immediately before SUDEP (Lacuey et al., 2016; Myers et al., 2018). In SUDEP, derangement of HRV might reflect abnormal central autonomic modulation, in which postictal generalized EEG suppression, an abnormal EEG pattern that has been reported to occur in almost all published SUDEP cases, might reflect altered brainstem function (Devinsky et al., 2016). Indeed, MRI studies show that progressive brainstem atrophy affecting cardiovascular autonomic control centres is associated with altered HRV (Mueller et al., 2018).

Sivathamboo et al. (2021) conducted a multicentre retrospective case–control study to compare HRV in 31 SUDEP cases and 56 matched living epilepsy control subjects using short-term interictal ECG recordings during wakefulness and sleep. The time- and frequency-domain components of HRV that were calculated included low- (0.04–0.14 Hz) and high-frequency power (0.15–0.40 Hz). High-frequency power is thought to represent vagal cardiac tone, whereas low-frequency power is considered to be a reflection of sympathetic and vagal cardiac tone (Shaffer & Ginsberg, 2017). The results from Sivathamboo et al. (2021) showed that increased high-frequency power during sleep was associated with a longer duration of survival in SUDEP cases. Given that high-frequency power is thought to represent vagal outflow to the heart, the finding by Sivathamboo et al. (2021) suggests that increased cardiovagal tone might be cardioprotective in SUDEP. Importantly, this might underlie the mechanism by which VNS treatment might be associated with low SUDEP rates seen in drug-refractory patients treated with VNS, of which these effects appear to increase with time (Ryvlin et al., 2018).

Antiseizure medications are the mainstay of epilepsy treatment. Despite the introduction of multiple new antiseizure medications over the last 30 years, one-third of all epilepsy patients will not achieve seizure freedom (Chen et al., 2018). Drug-resistant epilepsy is defined as failure of adequate trials of two (or more) tolerated, appropriately chosen and used antiseizure medication regimens to achieve freedom from seizures (Kwan et al., 2011). In patients with drug-resistant focal epilepsy, epilepsy surgery to remove or disconnect epileptogenic brain tissue can be considered in eligible candidates and provides the highest chance of seizure freedom. In those with drug-resistant epilepsy who are not eligible for surgery, dietary therapies, such as the ketogenic diet, or neurostimulation-based therapies including VNS, deep brain stimulation or responsive neurostimulation (RNS), can be considered on a case-by-case basis.

Potential mechanisms by which VNS influences physiological function include alteration of neurotransmitter release, changes to regional cerebral blood flow, synchronization or desynchronization of cortical and subcortical rhythms, and changes in functional connectivity between different brain regions (Carron et al., 2023).

**Origins of VNS in epilepsy.** The concept that electrical stimulation of the vagus nerve could have therapeutic effects in patients with epilepsy has existed for a considerable time. In the late 19th century, the American neurologist James Leonard Corning hypothesized that seizures were caused by an increase in blood flow to the brain and reported that manual compression of the carotid artery and external stimulation of what he believed to be the vagus nerve (although he was probably acting simply on the carotid arterial baroreceptors) simultaneously diminished seizures in epileptic individuals (Corning, 1883; Lanska, 2002). Although Corning's technique seemed promising, the results of his investigations were

not easy to interpret, and the method he implemented to treat epilepsy patients became unpopular after his death in 1923 (Badran & Austelle, 2022). It was also unclear whether or not vagal stimulation affected cortical function via direct afferent projections. Bailey and Bremer (1938) endeavoured to address this question by analysing cortical activation patterns in cats. They showed that VNS induced activation of the orbital surface of the frontal cortex, as measured by an increased number of electrical potentials recorded by electrodes placed on the surface of the cortex. The implication of their findings was that the vagus nerve contained afferent projections to cortical regions that, when stimulated, could modulate brain activity. Zanchetti and colleagues (1952), using a similar isolated brain model to Bailey and Bremer (1938), showed that electrical stimulation of the cervical vagus nerve in spinalized cats could suppress spontaneous EEG activity and reduce strychnine-evoked cortical spikes. EEG is an important clinical tool because it measures brain activity using electrodes placed either on the surface of the scalp or intracranially, and it can be used to help determine regions of the brain involved in epileptic seizures (Feyissa & Tatum, 2019). This was the first study with evidence suggesting that VNS could eradicate or transiently prevent hyperexcitability or synchronization of cortical activity. Following on from this finding, Chase et al. (1966, 1967) used VNS in the cat to investigate correlations between electrical activity recorded from the vagus nerve and the induced cortical and subcortical responses. The researchers found that VNS seemed to induce cortical synchronization and desynchronization, and these responses were attributed to structurally and functionally distinct fibre groups that compose the vagal afferent projections to the cortex (Chase et al., 1966).

Some 30 years later, Zabara (1985) was able to attenuate seizures in dogs via electrical stimulation of the cervical vagus. At this time, stimulation of the vagus nerve showed its true potential as a therapeutic mechanism for the treatment of epilepsy. A few years later, the first VNS device was first implanted in humans. Of the four patients implanted, two became completely seizure free, one had ∼40% reduction in seizure frequency, and the other had no improvement in seizure control (Penry & Dean, 1990).

Subsequent pilot studies (Uthman et al., 1990; Wilder et al., 1991) and the results from randomized controlled trials (Ben-Menachem et al., 1994; George et al., 1994; Handforth et al., 1998) confirmed the efficacy of VNS for the treatment of drug-refractory epilepsy. Based on the results of these early trials, in 1997 the US Food and Drug Administration approved VNS as an adjunct treatment option for patients with drug-refractory epilepsy (also referred to as drug-resistant epilepsy) (Groves & Brown, 2005).

**Clinical aspects of VNS in epilepsy.** VNS has been an approved therapy in Europe since 1994 and in the USA since 1997 for the treatment of drug-resistant epilepsy (Nemeroff et al., 2006). In 2001, VNS was also approved as a therapy for patients with treatment-resistant depression and bipolar disorder in Canada and Europe. In 2005, the US Food and Drug Administration also approved VNS for the treatment of patients with treatment-resistant depression. Currently, VNS is an approved therapy for patients with drug-resistant epilepsy and treatment-resistant depression in the USA, UK and Europe. In Australia, VNS was approved by the Therapeutic Goods Administration in 2000 to treat patients with drug-resistant epilepsy (Yates et al., 2022), its only approved use.

The VNS pulse generator, roughly the size of a cardiac pacemaker, is implanted subcutaneously in the anterior wall of the chest, as shown in Fig. 2. The bipolar stimulating electrodes are wrapped around the left cervical vagus nerve through an incision in the neck, then both components are connected via a tunnelling procedure (Agnew & McCreery, 1990; Nemeroff et al., 2006). The rationale for targeting the left vagus nerve for stimulation in the context of epilepsy is that it does not innervate the SA node of the heart and therefore is anticipated to have less effect on heart rate (Carreno & Frazer, 2017; Kunze, 1972). In the treatment of epilepsy, there have been only two case reports of late-onset bradycardia and asystole associated with left-sided VNS (Pascual, 2015; Shankar et al., 2013).

The VNS device is linked to a telemetric wand that allows clinicians to adjust stimulation parameters, including the output current, pulse width, frequency and the duty cycle. Clinical VNS systems typically deliver intermittent stimulation, and the duty cycle refers to the amount of time for which the device is on or off (Nemeroff et al., 2006). Stimulation is usually initiated 2 weeks after implantation, and stepwise titration of output current (usually every 2 weeks) occurs over ∼3 months to achieve an output current of 1.5–2 mA (Fahoum et al., 2022). Other parameters are standardized and usually held constant in this phase; pulse width of 250 μs, frequency of 20 Hz and duty cycle of 10% are recommended by the manufacturer, for example. These settings are derived from the pivotal randomized controlled trials (Ben-Menachem et al., 1994; Handforth et al., 1998) and subsequent studies demonstrating equivalency. Alternative programming strategies have been suggested for when standardized programming fails to achieve the desired therapeutic response (Haddad et al., 2023); however, there has been no systematic study to identify optimal parameters for clinical stimulation, hence only general guidelines exist. Indeed, optimal stimulation

parameters are likely to vary individually, hence aiming to identify a single ideal parameter set for all patients might be fruitless. In practice, after the initial titration phase, VNS parameters are adjusted individually, according to the therapeutic response and tolerability of adverse effects.

Most adverse effects seen in refractory epilepsy patients who undergo insertion of a VNS device range from difficulty in speaking (dysphonia) and alterations in voice to coughing during the period of stimulation (Johnson & Wilson, 2018). Up to 66% of VNS-implanted patients will experience some form of voice change, typically hoarseness during stimulation (Ben-Menachem, 2002; Handforth et al., 1998). Other VNS-related side-effects, including headache, pain, dyspnoea, cough and paraesthesia, are present in ∼16–25% of patients at 3 months. At 12 months, these numbers decline to effect ∼13–16% of patients and become rare at the 5 year mark, affecting only 1–5% of patients (Ben-Menachem, 2002). One other side-effect observed clinically in VNS patients is sleep-disordered breathing and the exacerbation of obstructive sleep apnoea, typically managed by alternate day/night setting adjustments made by clinicians (Simpson et al., 2022).

**Fibre activation.** Excitation of fibres within the vagus nerve depends on multiple factors relating to the nerve itself; these include the fibrous tissue around the nerve fascicles, myelination of nerve fibres and fibre diameter (Stakenborg et al., 2020). Placement of the electrode cuff used to stimulate the nerve can also influence which fibres are activated during stimulation. Further to this, following surgical implantation of the cuff electrodes, fibrous tissue formation around the electrodes can increase resistance and require higher stimulation parameters for activation (Helmers et al., 2012). All these factors might be a reason why clinically there are differences in patient responses to vagal nerve stimulation in the context of drug-resistant epilepsy.

Understanding the morphology of the vagus nerve is key to tailoring stimulation parameters to maximize the effectiveness of device therapy whilst also mitigating side-effects. Other factors that influence fibre activation thresholds include the diameter of vagal fascicles and their organization. In their anatomical investigations of the cervical vagus nerve in humans, pigs and mice, Stakenborg et al. (2020) concluded that the increased size of the nerve was associated with increased fibrous tissue (perineurium) surrounding fascicles and a larger number of fascicles in pigs and humans in comparison to mice. These two factors are known to increase the activation threshold of medium- and large-diameter myelinated fibres, meaning that a higher current intensity is necessary for activation of these fibres in humans and pigs. Likewise, Pelot et al. (2020) quantified the morphology of the human, porcine and rat vagus nerve to confirm that the diameter was sub-

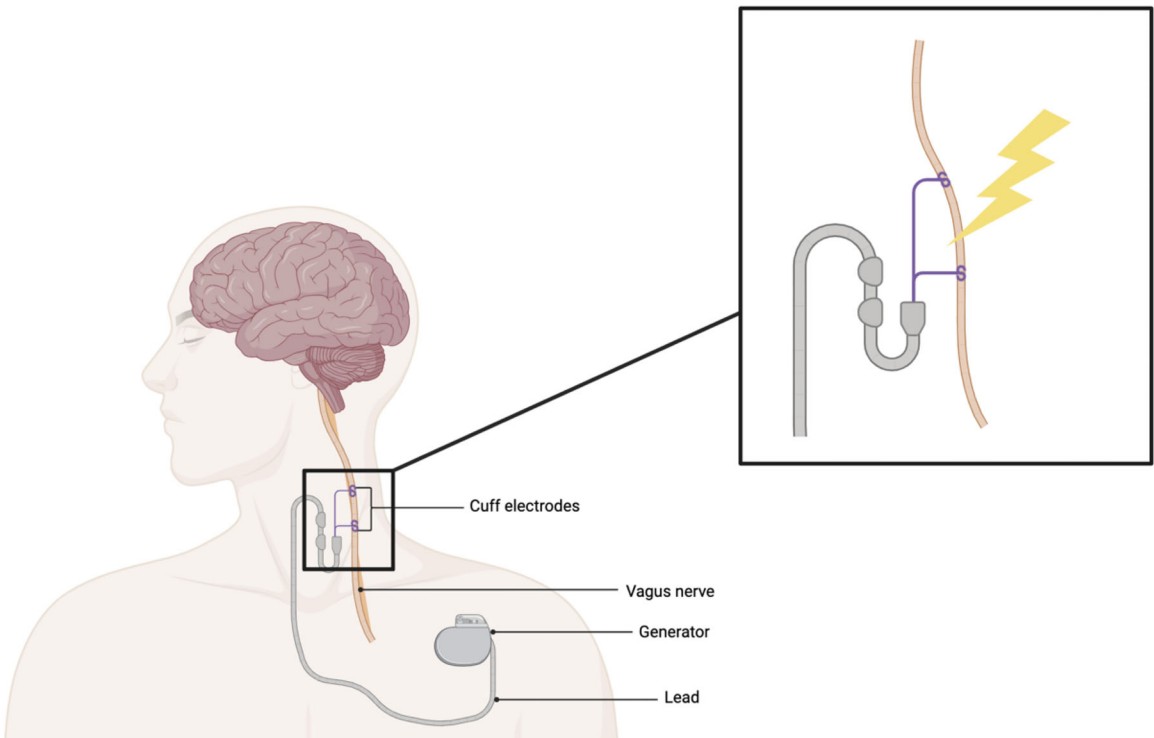

**Figure 2. Schematic diagram of an implanted vagus nerve stimulator device**
Figure created with BioRender.

stantially different in humans and pigs compared with rats (Fig. 3) and that perineurium thickness differed across species (Fig. 4). These investigators also showed that although the porcine vagus was of a similar size to the human vagus, it had a greater number of fascicles, and these were smaller in size. Thus, investigations of VNS parameters in animal models should be considered carefully concerning clinical implications, because their vagal anatomy differs significantly from humans. Attempts at optimizing stimulation parameters (discussed in more detail later) in other species might not reflect what is needed for VNS applications in humans.

With the advent of computational modelling, researchers have become able to simulate activation thresholds of vagal fibres based on morphological characteristics of the nerve (Musselman et al., 2023). Davis et al. (2023) used realistic anatomical models of the cervical vagus nerve to show that larger-diameter fascicles had higher activation thresholds and attributed this to the presence of a thicker perineurium. Larger fascicles with a thicker perineurium have increased resistance, meaning that the current is effectively blocked from entering and activating fibres. The opposite was true for fibre activation in smaller fascicles, the assumption being that the thickness of the perineurium and the diameter of the endoneurium are smaller, resulting in lower activation thresholds. These investigators also showed that fascicle diameter had a greater impact on activation thresholds than fascicle location (the distance between the stimulating electrode and fascicle) (Davis et al., 2023).

**Stimulation parameters and seizure reduction in epilepsy.**
There is uncertainty surrounding the optimal stimulation parameters for VNS, including knowledge of which fibres

are being activated by which parameter sets, and how this might explain the variability seen in treatment response. Clinical stimulation parameters include output current, duty cycle (stimulation on- and off-time), pulse width and frequency. These parameters are titrated algorithmically for all patients initially. If treatment response is inadequate, parameters are then varied individually for each patient to achieve effective seizure control and avoid side-effects. There is comparatively little evidence to guide clinicians during this phase of programming. A better understanding of fibre activation, the effect of differing stimulation parameters, and the relationship of both to clinical outcomes would be beneficial.

Multiple human studies have demonstrated the efficacy of VNS as an adjunct treatment for epilepsy. In a study of 195 patients, DeGiorgio et al. (2000) demonstrated that high-frequency VNS (30 Hz) resulted in a median decrease of seizure frequency of 34% at 3 months post-VNS implantation, compared with the post-implantation baseline, which increased to 46% at 12 months. Responses to VNS therapy observed at 3 months post-implantation with the device tend to increase for a duration of 2 years; thereafter, response to treatment levels off (Morris & Mueller, 1999). However, several studies have shown that the efficacy of VNS in seizure reduction increases over time. At 3 months post-VNS implantation, between 23 and 31% of patients respond to (i.e. experience ≥50% reduction in seizure frequency) VNS therapy (Ben-Menachem et al., 1994; Handforth et al., 1998), which increases to 31–49% of patients at the 1 year mark (DeGiorgio et al., 2000; Englot et al., 2016; Morris & Mueller, 1999; Salinsky et al., 1996). Little information is available with regard to long-term follow-up of seizure freedom rates amongst patients. However, retrospective analysis from one patient registry

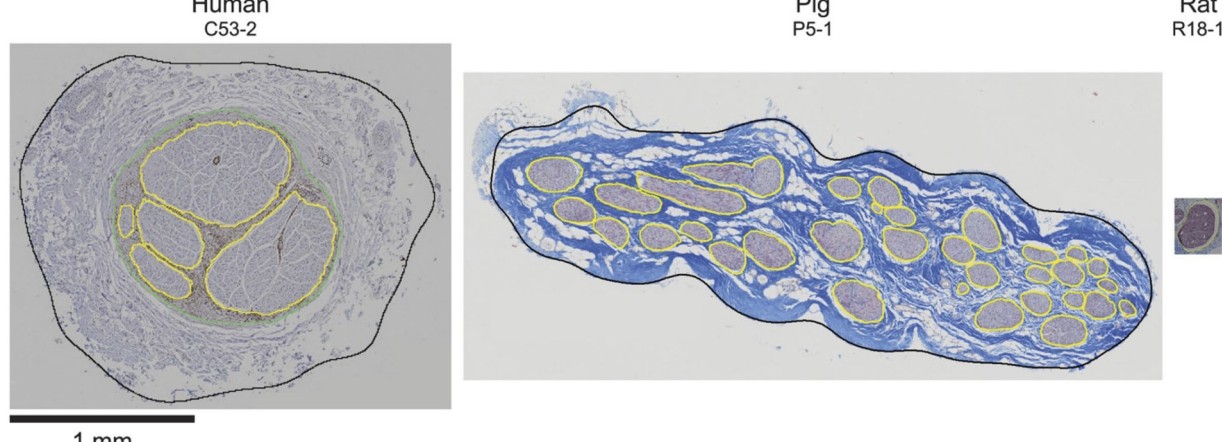

**Figure 3. Cervical vagus nerve transverse sections**
Cervical vagus nerve transverse sections at the same scale from the human, pig and rat. Reproduced with permission from Pelot et al. (2020).

has shown seizure freedom rates of 8% with VNS (Englot et al., 2016). There are also a small number of patients (3–4%) who worsen, experiencing a >50% increase in seizure frequency, with VNS therapy (1995; DeGiorgio et al., 2000). Two years after the device is switched on, patients will see a median seizure reduction of 44–58% (Elliott et al., 2011; Morris & Mueller, 1999).

Performing sham (or placebo)-controlled studies involving VNS is challenging owing to the obvious sensations associated with initiation of VNS, even at low settings (hoarseness, coughing and a sensation of stimulation). Hence, investigations into the efficacy of VNS in drug-resistant epilepsy patients have compared 'high' and 'low' stimulation parameter groups. High stimulation parameters for VNS generally refer to a frequency of 30 Hz, a duty cycle of 30 s on and 5 min off, and a pulse width of 500 μs. Output currents in these trials were titrated to tolerance over several weeks,

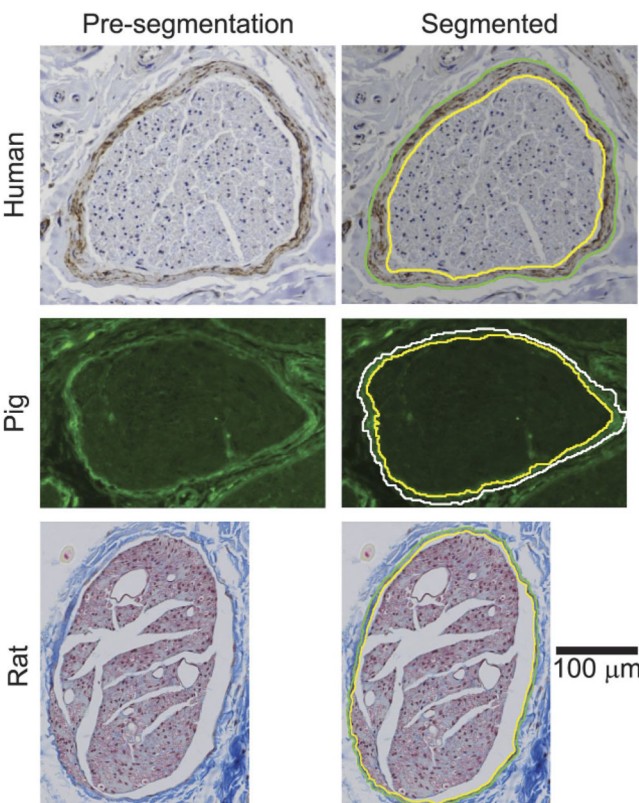

**Figure 4. Perineurium thickness of fascicles in the human, pig and rat vagus nerve**
Perineurium thickness of fascicles in the human, pig and rat vagus nerve assessed using anti-claudin-1 immunohistochemistry for human nerves, anti-fibronectin immunofluorescence for pig nerves, and Masson's trichrome histology for rat nerves. Pre-segmentation images show raw segments, contrasting the perineurium with surrounding tissue. Segmented images show segmented labelling of the inner and outer boarders of the perineurium. The rat perineurium is thinnest and the human perineurium is thickest. Reproduced with permission from Pelot et al. (2020).

typically to 1.3–1.5 mA, but ≤3.5 mA. Low stimulation parameters usually indicated a frequency of 1–2 Hz, a duty cycle of 30 s on and 60–180 min off, and a pulse width of 130 μs. Output currents were generally lower and titrated to sensory perception threshold, but then not increased further. Ben-Menachem et al. (1994) studied 67 patients over the course of 14 weeks who were randomized to receive either high or low VNS. Although both stimulation protocols were effective at reducing seizure frequency in epilepsy patients, a greater mean seizure frequency reduction was observed with high stimulation (~30%) compared with low stimulation (~11%) (Ben-Menachem et al., 1994). Similar results were obtained by Handforth et al. (1998), who compared the high and low stimulation in a larger patient cohort. High stimulation led to a 28% average reduction in seizure frequency for patients, whereas low stimulation led to a 15% reduction in seizure frequency on average.

Although the 'low' stimulation settings are considered not to be clinically effective and/or to be a placebo, these studies note a small but appreciable benefit. Programming has evolved somewhat over time, and now the most typical parameters used are referred to as standard settings and are a frequency of 20–30 Hz, pulse duration of 250–500 μs, with 30–60 s of stimulation on-time and 1.1–5 min off-time (Groves & Brown, 2005).

**Mechanisms of action.** In an effort to understand the underlying mechanism of VNS therapy, research into the effects of VNS at an electrophysiological level has been undertaken (Larsen et al., 2016a, b; Nichols et al., 2011). Preclinical work on the effects of VNS on hippocampal activity has highlighted that a rapid duty cycle (7 s on, 18 s off) is more effective at reducing hippocampal power at lower current outputs (250 μA) than a standard cycle (30 s on, 5 min off) at high current outputs (1000 μA) (Larsen et al., 2016a). Their study also revealed that reductions in hippocampal power during rapid-cycling VNS reached a saturation at ~500 μA, suggesting that higher current outputs are not needed to produce maximal electrophysiological effects. Phase–amplitude coupling (PAC) is a measure of synchronization of different local neuronal populations in the brain and has important implications in epilepsy (Edakawa et al., 2016). It has been shown that PAC between theta and gamma frequency rhythms was markedly reduced during rapid-cycle VNS at 250 μA (Edakawa et al., 2016). Edakawa et al. (2016) showed that PAC during the ictal period (time during a seizure) was stronger than during the time between seizures (interictal period) in drug-refractory epilepsy patients. Thus, the ictal suppression of PAC might be responsible for part of the antiseizure effects of VNS and might be an important biomarker of improved seizure control. Rapid-cycling VNS has been shown to reduce PAC in healthy rodent

brains (Larsen et al., 2016a). Hence, rapid cycling might represent a specific way to maximize the benefit of PAC suppression in humans, and at a lower than normal output current; and at the same time improve efficacy, tolerability and battery life.

Preclinical studies have also laid the foundations for our understanding of the brain regions involved in the anti-epileptic effects of VNS. Early animal work by Naritoku et al. (1995) quantified the expression of fos, a protein marker for elevated neuronal activity, in rats to determine which brain regions are implicated in the anti-convulsant effects of VNS. Animals exposed to VNS (30 Hz, 500 μs, 30 s on, 5 min off) showed activation of several limbic structures, including the amygdala, the diencephalon (including the habenula and lateral posterior thalamic nuclei), ventromedial and arcuate hypothalamic nuclei, and in the brainstem, including the LC. Human PET studies have also shown activation of these regions during VNS at clinically relevant parameters (Henry et al., 1998). Henry et al. (1998) demonstrated in epilepsy patients that acute VNS of the left cervical vagus nerve results in increases in regional blood flow in regions that are known to receive vagal input and generate vagal output, such as the rostra1 medullary region (which contains the NTS and DMNV), hypothalamus, thalamus, anterior insula, orbitofrontal cortex, inferior frontal gyrus and inferior parietal pole, with decreases in blood flow in the hippocampus, amygdala and posterior cingulate gyrus. In humans, functional imagining studies have shown a clinical heterogeneity in the brain regions that appear to be activated or suppressed during VNS (Chae et al., 2003). This might be attributable to small sample sizes, varying epilepsy types and aetiologies, differing duration of disease, varying duration of treatment with VNS and varying antiseizure medications. Perhaps the strongest factor impacting the clinical variance in the activation and suppression of higher-order brain regions is the time since implantation and activation of VNS devices at which these brain imaging studies have taken place (Chae et al., 2003). Functional imaging performed on individuals who have had chronic VNS therapy (months to years) show altered brain functionality when compared with patients who are imaged almost immediately post-initiation of VNS. The implication here is that after a period of time, VNS results in changes in activity within different brain regions. Further imaging investigations in a clinical setting are required at different time points post-VNS implantation to track any changes in neuronal activity that might result with chronic stimulation therapy.

**The role of fibre type in VNS for epilepsy.** Early preclinical work in canines suggested that VNS attenuation of seizures was mediated by the activation of unmyelinated C fibres, which make up the majority of the sensory component of the cervical vagus nerve (Woodbury & Woodbury, 1990). The authors argued that high stimulation levels, which would, in theory, recruit the majority of C fibres, was required for the antiseizure effect of VNS. More recent investigations, however, have shown that C fibres might not be necessary for seizure control. Krahl et al. (2001) showed that destruction of C fibres by capsaicin in rats did not attenuate the anti-epileptic effects of VNS. This evidence is also supported by Koo et al. (2001), who stimulated the vagus in a cohort of drug-refractory epilepsy patients under anaesthesia after inserting a VNS device. They concluded that the compound nerve action potential evoked by stimulation was characteristic of A and B fibres and that the stimulus currents used were not high enough to recruit C fibres. Likewise, intraneural vagal recordings by Evans et al. (2004) showed predominantly A fibre-type responses to stimulation of the cervical vagus in epilepsy patients under anaesthesia. Notably, C fibre responses to cervical vagus stimulation were also recorded in some patients (Evans et al., 2004). Fibre activation thresholds informed by electrical stimulation of the human vagus nerve intraoperatively and in awake epilepsy patients with implanted VNS devices are summarized in Fig. 5.

Clinical stimulation settings for drug-resistant epilepsy patients include current intensities of 1–3.5 mA (although 1.5–2 mA is typical), a stimulation frequency of 20–30 Hz and pulse widths of 250, 500 or, less frequently, 130 μs, combined with different cycling times (Fahoum et al., 2022). Research in humans has shown that stimulation of the vagus nerve at pulse widths of 500 and 300 μs can result in equal activation levels, but decreasing the pulse width to 100 μs required a higher stimulus current in order to activate the nerve to the same extent (Koo et al., 2001). This indicates that it might not be beneficial to decrease the pulse width to the lowest setting of 130 μs unless output current is increased to compensate. Therefore, it seems that stimulating at a pulse width of 250 μs is not only tolerable for patients but is also more likely to conserve battery life (Koo et al., 2001).

It is difficult to determine which fibres are stimulated by VNS in patients in routine clinical care, given the difficulty in performing the required recordings in this setting and the varying current intensities (amplitudes) and pulse widths that are required to activate A, B and C fibres. The above study was limited in its ability to test fibre activation under anaesthesia owing to patient safety. Thus, there is much to be learned in recording from patients already implanted with VNS devices who are awake and who have different clinical stimulation settings, which would allow for a range of current amplitudes to be tested, because some patients will have much stronger settings than others. Recently, we performed the first microelectrode recordings from the cervical vagus nerve

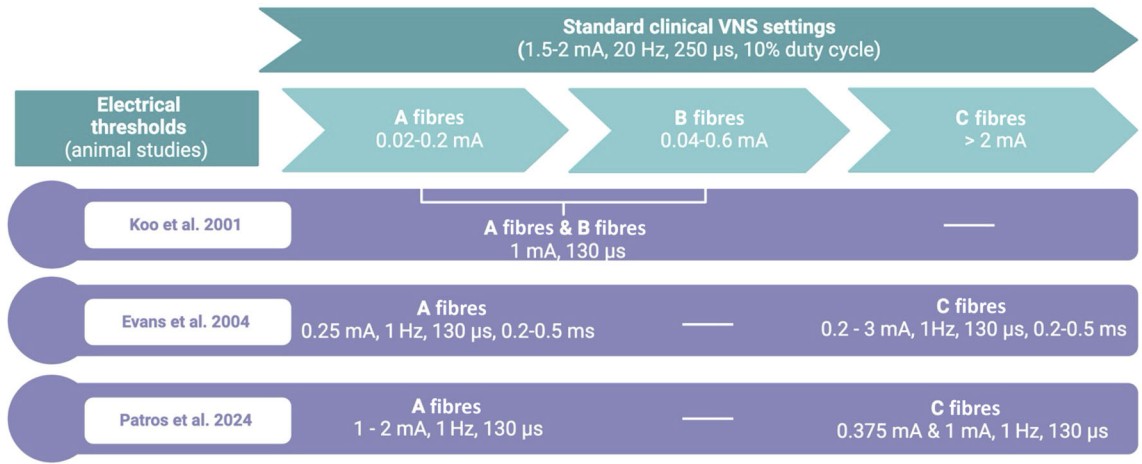

**Figure 5. Vagal fibre activation thresholds**
Summary of vagal fibre activation thresholds for A, B and C fibres from electrical stimulation of the human cervical vagus nerve in awake (Patros et al., 2024) and anaesthetized epilepsy patients (Evans et al., 2004; Koo et al., 2001). Created with BioRender.

in two epilepsy patients with implanted VNS devices and showed that activation of both unmyelinated and myelinated axons can be achieved at current intensities of <1 mA (Patros et al., 2024).

Little is known about the effects of VNS on the function of other organ systems in the body in the context of epilepsy. Koopman et al. (2016) investigated the production of tumour necrosis factor, a therapeutic target in the treatment of autoimmune conditions, in a cohort of epilepsy patients. Patients underwent transient stimulation of the cervical vagus under general anaesthesia, for a single 30 s train (20 Hz, 500 µs and 1 mA). Despite this brief stimulation, tumour necrosis factor production, along with production of the inflammatory molecules interleukin-6 and interleukin-1$\beta$, was inhibited. These data support the involvement of the vagus nerve in inflammatory signalling in patients without a history of inflammatory disease. Of interest is the level of stimulation required to produce these anti-inflammatory effects, particularly given that the vagal innervation of the gut is provided by C fibres, which, as noted above, have much higher electrical thresholds than myelinated axons. Interestingly, modulation of inflammatory signalling was achieved with only transient stimulation, as opposed to the chronic stimulation used clinically in drug-refractory epilepsy patients to achieve seizure control.

## Conclusion

The vagus nerve is the largest cranial nerve, innervates most organs in the thoracic and abdominal cavities, and plays a vital role in homeostatic functions of diverse body systems through its mixed composition of afferent and efferent fibres. The anatomical location of the vagus nerve means that it is a convenient and accessible target for stimulation-based therapies. Together, these features make it a powerful tool to interact with and treat a variety of disease states in humans, most notably neurological disorders, such as epilepsy. A better understanding of the effects of VNS in humans is required to optimize outcomes with VNS, and to do this better tools are required to explore and quantify sympathetic and parasympathetic effects and vagal tone in general.

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

## Additional information

### Competing interests

M.P. reports no disclosures. S.S. is the recipient of a National Health and Medical Research Council Investigator Award (APP2025610). She is supported by Research Program Grants from the National Institute of Health (1U54AT012307-01 and 1R01NS123928-01). She reports salary support paid to her institution from Jazz Pharmaceuticals for clinical trial-related activities; she receives no personal income for these activities. H.D.S. is supported by a National Health and Medical Research Council (NHMRC) Medical Research Future Fund Grant (MRFF 2025695). He has received travel support for educational purposes and reports consulting fees to his institution from LivaNova. T.J.O'B. is supported by a Program Grant (APP1091593) and Investigator Grant (APP1176426) from the National Health and Medical Research Council of Australia. He reports grants and consulting fees to his institution from LivaNova, Eisai, UCB Pharma, Praxis, Biogen, ES Therapeutics and Zynerba.

## Author contributions

M.P.: Drafting the work and revising it critically for important intellectual content; Final approval of the version to be published; Agreement to be accountable for all aspects of the work. S.S.: Revising the work critically for important intellectual content; Final approval of the version to be published; Agreement to be accountable for all aspects of the work. H.D.S.: Revising the work critically for important intellectual content; Final approval of the version to be published; Agreement to be accountable for all aspects of the work. T.J.O'B.: Revising the work critically for important intellectual content; Final approval of the version to be published; Agreement to be accountable for all aspects of the work. V.G.M.: Conception or design of the work; Revising the work critically for important intellectual content; Final approval of the version to be published; Agreement to be accountable for all aspects of the work. All authors approved the final version of the manuscript, all persons designated as authors qualify for authorship, and all those who qualify for authorship are listed.

## Funding

This was work was supported by funding from the National Institutes of Health to V.G.M. and T.J.O'B. (1U54AT012307-01) and from the National Health and Medical Research Council of Australia to V.G.M. (GTN 2 019 404).

## Acknowledgements

Open access publishing facilitated by Monash University, as part of the Wiley - Monash University agreement via the Council of Australian University Librarians.

## Keywords

autonomic control, epilepsy, fibre activation, vagus nerve, vagus nerve stimulation

## Supporting information

Additional supporting information can be found online in the Supporting Information section at the end of the HTML view of the article. Supporting information files available:

**Peer Review History**

