## [Peer Review History · The Journal of Physiology]

The physiology, anatomy and stimulation of the vagus nerve in epilepsy

Mikaela Patros, Shobi Sivathamboo, Hugh David Simpson, Terence J O'Brien, and Vaughan G Macefield
DOI: 10.1113/JP287164

Corresponding author(s): Vaughan Macefield (vaughan.macefield@monash.edu)

The following individual(s) involved in review of this submission have agreed to reveal their identity: Rohit Ramchandra (Referee #1); Emma C Hart (Referee #2)

Review Timeline:

Submission Date:	02-Sep-2024
Editorial Decision:	01-Oct-2024
Revision Received:	22-Dec-2024
Accepted:	12-Feb-2025

Senior Editor: Laura Bennet

Reviewing Editor: Christopher Lear

Transaction Report:

Dear Vaughan,

Re: JP-TR-2024-287164 "The physiology, anatomy and stimulation of the vagus nerve" by Mikaela Patros, Shobi Sivathamboo, Hugh Simpson, Terence J O'Brien, and Vaughan G Macefield

Thank you for submitting your manuscript to The Journal of Physiology. It has been assessed by a Reviewing Editor and by 2 expert referees and we are pleased to tell you that it is acceptable for publication following satisfactory revision.

ABSTRACT FIGURES: Authors may use The Journal's premium BioRender account to create/redraw their Abstract Figures (and any other suitable schematic figure). Information on how to access this account is here: <https://physoc.onlinelibrary.wiley.com/journal/14697793/biorender-access>.

REVISION CHECKLIST: Upload a full Response to Referees file. To create your 'Response to Referees' copy all the reports, including any comments from the Senior and Reviewing Editors, into a Microsoft Word, or similar, file and respond to each point, using font or background colour to distinguish comments and responses and upload as the required file type.

We look forward to receiving your revised submission.

Best wishes,

Laura Bennet
Senior Editor

EDITOR COMMENTS

Reviewing Editor:

Thank you for your well written topical review. It has been well received by our two reviewers, please attend to their comments. Both reviewers have suggested that a deeper discussion on the physiology through which VNS mediates its beneficial effects on epilepsy would be beneficial (i.e. the comment re afferent vs efferent fibers from reviewer 1, and the physiological meaning of the VNS parameters from reviewer 2).

Please also see 'Required Items' below.

Additionally:

- 1) I agree that a title including epilepsy would be appropriate to attract the correct audience for this review.
- 2) The review would be aided by an improved abstract that includes key take home messages found in the review rather than simply stating the topics.
- 3) An appropriate abstract figure that includes concepts found in the review needs to be included, rather than the current simple illustration.
- 4) Please consider further accompanying explanatory/summary diagrams (or illustrative data as appropriate) to enhance the rest of the review.

REFEREE COMMENTS

Referee #1:

This is a comprehensive review detailing the anatomy of the vagus nerve and a second part highlighting stimulation of the vagus nerve as a treatment for epilepsy. Overall, this is well-written and the initial detail of the anatomy to help understand how VNS works in epilepsy is welcome. I have a few comments about the manuscript.

1 - Lines 191-198 and line 407. I think the reference (Kunze in 1972) is more a description of the changes in cardiac vagal single unit activity and does not compare stimulation of the right vs. the left vagus. Please check this reference. Another point to make here is that the commonly mentioned fact that stimulation of the left cervical vagus does not alter heart rate has not been confirmed by other studies. A number of labs using different preparations have shown a reproducible decrease in HR with left vagus nerve stimulation and this needs to be mentioned somewhere.

e.g. Brack KE, Coote JH, Ng GA. *Exp Physiol.* 2010 Jan;95(1):80-92. doi: 10.1113/expphysiol.2009.048215.

2- Line 354 - Epilepsy, mortality and drug-resistance.

I think this paragraph should perhaps be at the start of the Epilepsy sections. The comments on VNS in epilepsy seems to come before understanding why this was trialled and why this was needed in patients with epilepsy. I suggest the flow might be better with this section earlier.

3 - The role of fibre type in VNS for epilepsy section suggests that the benefits of VNS may come from A and B fibre stimulation. This suggests that the effects are not necessarily mediated by afferent stimulation. Any suggestion on what organs may be targeted to cause the responses? In the same vein, given the extensive details presented about the innervation of the heart, pulmonary circulation and the gastrointestinal tract by the vagus in the first section, I wonder if some comment could be made on what changes (if any) are observed in these organ functions during VNS for epilepsy. Anything to tie back to the first section would be useful to readers.

Typographical comments:

Line 56 - perhaps reword this. Descends along its path suggests there was a path for it to follow.

Line 260 - given the sympathetic and parasympathetic nerves have activity along these nerves, perhaps best to not use cardiac activity in this sentence.

Line 305 - Its use has been the best studied.....

Line 415 - the device is on OR off

Referee #2:

The review by Patros et al aims to examine what can be learned about the effects of vagal nerve stimulation when VNS is done in patients with drug resistance epilepsy. Overall this a very well and carefully written review. We understand so much less about the vagus nerve in humans than we do about sympathetic control. With the increased use and interest surrounding vagal nerve stimulation as a treatment for many conditions and diseases - we need to understand more about the vagus. Thus this review is important.

Some things to consider:

The review is obviously very much focused on epilepsy - could the authors add this to the title?

Overall I think there needs to be more discussion about what the VNS parameters in humans means physiologically. The authors review some of this in separate sections; but how do the frequencies of the stimulation, for example, reflect actual input into the brainstem? How would this impact the normal firing rate of cells in the NTS for example? Does this stimulation override important afferent feedback?

I also think the manuscript could be improved by more discussion over why there are decreases in SUDEP with VNS - what can we learn from this? Are there less deaths due to arrhythmia for example? Linked to this I think it would also be beneficial to talk more about how the VNS impacts control of the heart and sympathetic innervation. With this in mind, does slowing of AV conduction with VNS increase the risk of long QT intervals?

Line 406 - if the left vagus nerve is targeted for VNS - then why would the authors anticipate it has an effect on HRV - when the SA node seems to be responsible for driving the variability? Perhaps some comment on this somewhere would be helpful. Interestingly - atropine does not have an affect on AV conduction. <https://pubmed.ncbi.nlm.nih.gov/7733345/>

Minor comments:

Intro - spleen important organ innervated by the vagus - potentially add this?

Line about GVA (108) innervating aortic chemoreceptors - isn't this also true for the carotid chemoreceptors and baroreceptors?

Line 135 - define NA and DMNV

Line 131 - afferent projections of the vagus - shouldn't this section be vagal afferent network within the brain
(otherwise people might expect this section to be about afferent innervation of other organs)

Line 198 - Vagus also promotes lusitropy?

Line 237 - also the carotid chemoreceptors

REQUIRED ITEMS

- Please include an Abstract Figure file, as well as the Figure Legend text within the main article file. The Abstract Figure is a piece of artwork designed to give readers an immediate understanding of the Review Article and should summarise the main conclusions. If possible, the image should be easily 'readable' from left to right or top to bottom. It should show the physiological relevance of the Review so readers can assess the importance and content of the article. Abstract Figures should not merely recapitulate other figures in the Review. Please try to keep the diagram as simple as possible and without superfluous information that may distract from the main conclusion of the Review. Abstract Figures must be provided by authors no later than the revised manuscript stage and should be uploaded as a separate file during online submission labelled as File Type 'Abstract Figure'. Please ensure that you include the figure legend in the main article file. All Abstract Figures will be sent to a professional illustrator for redrawing and you may be asked to approve the redrawn figure before your paper is accepted.

- Please upload separate high quality figure files via the submission form.

- Author profile(s) must be uploaded via the submission form. Authors should submit a short biography (no more than 100 words for one author or 150 words in total for two authors) and a portrait photograph of the two leading authors on the paper. These should be uploaded and clearly labelled together in a Word document with the revised version of the manuscript. Any standard image format for the photograph is acceptable, but the resolution should be at least 300 DPI and preferably more. A group photograph of all authors is also acceptable, providing the biography for the whole group does not exceed 150 words.

- It is the authors' responsibility to obtain any necessary permissions to reproduce previously published material and to list these within the main article file. For information, please see: https://jp.msubmit.net/cgi-bin/main.plex?form_type=display_requirements#permissions.

END OF COMMENTS

EDITOR COMMENTS

Reviewing Editor:

Thank you for your well written topical review. It has been well received by our two reviewers, please attend to their comments. Both reviewers have suggested that a deeper discussion on the physiology through which VNS mediates its beneficial effects on epilepsy would be beneficial (i.e. the comment re afferent vs efferent fibers from reviewer 1, and the physiologically meaning of the VNS parameters from reviewer 2).

Please also see 'Required Items' below.

Additionally:

1) I agree that a title including epilepsy would be appropriate to attract the correct audience for this review.

We thank the Editor for their comments and consideration. We have now amended the title as follows:

“The physiology, anatomy and stimulation of the vagus nerve in drug-resistant epilepsy”

2) The review would be aided by an improved abstract that includes key take home messages found in the review rather than simply stating the topics.

We thank the Editor for their comment. We have now amended the abstract.

3) An appropriate abstract figure that includes concepts found in the review needs to be included, rather than the current simple illustration.

We thank the Editor for their comment. We have now rectified the abstract to include key concepts found within the review.

4) Please consider further accompanying explanatory/summary diagrams (or illustrative data as appropriate) to enhance the rest of the review.

We thank the Editor for their comment. We have now included further diagrams to enhance some of the key findings discussed in the review.

REFEREE COMMENTS

Referee #1:

This is a comprehensive review detailing the anatomy of the vagus nerve and a second part highlighting stimulation of the vagus nerve as a treatment for epilepsy. Overall, this is well-written and the initial detail of the anatomy to help understand how VNS works in epilepsy is welcome. I have a few comments about the manuscript.

1 - Lines 191-198 and line 407. I think the reference (Kunze in 1972) is more a description of the changes in cardiac vagal single unit activity and does compare stimulation of the right vs. the left vagus. Please check this reference. e.g. Brack KE, Cooté JH, Ng GA. *Exp Physiol.* 2010 Jan;95(1):80-92. doi: 10.1113/expphysiol.2009.048215.

We thank the reviewer for their comment, we have addressed this error and replaced the reference with the appropriate reference (line 197-199).

“Animal investigations have demonstrated that electrically stimulating the right cervical vagus nerve had a more profound effect on the chronotropic function of the SA node compared to stimulating the left vagus nerve (Ardell & Randall, 1986)”.

Another point to make here is that the commonly mentioned fact that stimulation of the left cervical vagus does not alter heart rate has not been confirmed by other studies. A number of labs using different preparations have shown a reproducible decrease in HR with left vagus nerve stimulation and this needs to be mentioned somewhere.

We thank the reviewer for their comment. The left vagus nerve predominantly innervates the AV node and the right vagus nerve principally innervates the SA node. In humans we do not see reductions in HR with clinical left-sided VNS both clinically and even during our own physiological experiments where we continuously measure heart rate and blood pressure during clinical VNS cycles in epilepsy patients. We have now included a brief discussion on what effects are seen with right and left-sided VNS and mention some animal studies that have shown reductions in HR with left-sided VNS (lines 195-211).

“Fibers within the right vagus nerve innervate the SA node and fibers within the left vagus nerve innervate the AV node (Cheng et al., 1999; Cheng & Powley, 2000)... When comparing right and left VNS in the rabbit, it was shown that right VNS was more effective at producing bradycardia than left VNS (Brack et al., 2004). It should be noted that however that in some animal models stimulation of the left vagus nerve has been shown to evoke bradycardia. Selective stimulation of the left vagus nerve bundle in rats, significantly reduced heart rate by a varying degree with changes in stimulation frequency, amplitude and pulse width (Gierthmuehlen & Plachta, 2016). In pigs, intraneural closed-loop cervical stimulation of the left vagus nerve resulted in substantial reductions of heart rate (Sevcencu et al., 2018). The electrodes used by Sevcencu et al. (2018) however, were sutured longitudinally through the nerve, most likely leading to selective activation of fibers displaying chronotropic cardiac modulation. It is important to mention that achieving bradycardia with clinical left-sided VNS is highly unlikely, given that stimulation is not

selective or intraneural and that the left vagus nerve innervates the AV node, not the SA node.”

2- Line 354 - Epilepsy, mortality and drug-resistance.

I think this paragraph should perhaps be at the start of the Epilepsy sections. The comments on VNS in epilepsy seems to come before understanding why this was trialled and why this was needed in patients with epilepsy. I suggest the flow might be better with this section earlier.

We thank the reviewer for their comment. This section has now been moved to the beginning of the second part of the review “*Stimulation of the vagus nerve in epilepsy*” (line 341-403).

3 - The role of fibre type in VNS for epilepsy section suggests that the benefits of VNS may come from A and B fibre stimulation. This suggests that the effects are not necessarily mediated by afferent stimulation. Any suggestion on what organs may be targeted to cause the responses? In the same vein, given the extensive details presented about the innervation of the heart, pulmonary circulation and the gastrointestinal tract by the vagus in the first section, I wonder if some comment could be made on what changes (if any) are observed in these organ functions during VNS for epilepsy. Anything to tie back to the first section would be useful to readers.

We thank the reviewer for their comment. We are acutely aware of the unknowns with respect to how VNS works. We do know that electrical stimulation will excite both afferent and efferent axons, both orthodromically and antidromically. What is not known is what the effects of VNS are on other organ systems. Indeed, this is the aim of a large NIH grant we are part of, in which the aim is to define the effects of VNS on cardiovascular function, including muscle sympathetic nerve activity, cardiac conduction, immune function and metabolic profiles. Unfortunately, none of these data are yet at hand. We have included some discussion on what is known about the effects of VNS on inflammatory signalling in epilepsy patients into the section titled “*The role of fiber type in epilepsy*” (lines 739-752).

“Little is known about the effects of VNS on the function of other organ systems in the body in the context of epilepsy. Koopman et al. (2016) investigated the production of tumor necrosis factor (TNF), a therapeutic target in the treatment of autoimmune conditions, in a cohort of epilepsy patients. Patients underwent transient stimulation of the cervical vagus under general anesthesia, for a single 30 s train (20 Hz, 500 μ s and 1 mA). Despite this brief stimulation, TNF production, along with production of inflammatory molecules interleukin (IL)-6 and IL-1 β was inhibited. This data supports the involvement of the vagus nerve in inflammatory signalling in patients without a history of inflammatory disease. Of interest is the level of stimulation required to produce these anti-inflammatory effects, particularly given that the vagal innervation of the gut is provided by C fibres which, as noted above, have much higher electrical thresholds than myelinated axons. Interestingly, modulation of inflammatory signalling was achieved with only transient stimulation, as opposed to the chronic stimulation used clinically in drug-refractory epilepsy patients to achieve seizure control.”

Typographical comments:

Line 56 - perhaps reword this. Descends along its path suggests there was a path for it to follow.

We thank the reviewer for their comment. We have amended the wording as follows (line 56):

‘...and descends along the neck with both the carotid artery...’

Line 260 - given the sympathetic and parasympathetic nerves have activity along these nerves, perhaps best to not use cardiac activity in this sentence.

We thank the reviewer for their comment. We have amended the wording as follows (line 291):

“The parasympathetic and sympathetic divisions of the autonomic nervous system have divergent influences on autonomic activity...”

Line 305 - Its use has been the best studied.....

We thank the reviewer for their comment. We have amended the wording as follows (line 336-339):

“Stimulation of the vagus extends to psychiatric, gastrointestinal, and immunologic disease (Rush et al., 2000; Koopman et al., 2016; Bonaz et al., 2017) however, this section of the review will focus on its clinical applications in the context of drug-resistant epilepsy.”

Line 415 - the device is on OR off

We thank the reviewer for their comment. We have amended the wording as follows (line 552):

“...the device is on or off...”

Referee #2:

The review by Patros et al aims to examine what can be learned about the effects of vagal nerve stimulation when VNS is done in patients with drug resistance epilepsy. Overall this a very well and carefully written review. We understand so much less about the vagus nerve in humans than we do about sympathetic control. With the increased use and interest surrounding vagal nerve stimulation as a treatment for many conditions and diseases - we need to understand more about the vagus. Thus this review is important.

Some things to consider:

The review is obviously very much focused on epilepsy - could the authors add this to the title?

We thank the reviewer for their comment. We have amended the title as follows:

“The physiology, anatomy and stimulation of the vagus nerve in drug-resistant epilepsy”

Overall I think there needs to be more discussion about what the VNS parameters in humans means physiologically. The authors review some of this in separate sections; but how do the frequencies of the stimulation, for example, reflect actual input into the brainstem? How would this impact the normal firing rate of cells in the NTS for example? Does this stimulation override important afferent feedback?

We thank the reviewer for their comment. It is well known that myelinated axons can follow stimulus trains of up to 1000Hz, unmyelinated up to 100Hz (these are way beyond those frequencies used clinically 20-30Hz). We have added a section on mechanisms of action which discusses the electrophysiological effects of different parameter settings seen with VNS. Please see lines 689-739:

“Mechanisms of action:

In an effort to understand the underlying mechanism of VNS therapy, research into the effects of VNS at an electrophysiological level have been undertaken (Nichols et al., 2011; Larsen et al., 2016a; Larsen et al., 2016b). Preclinical work on the effects of VNS on hippocampal activity has highlighted that a rapid duty cycle (7 s on, 18 s off) is more effective at reducing hippocampal power at lower current outputs (250 μ A) than a standard cycle (30 s on, 5 min off) at high current outputs (1000 μ A) (Larsen et al., 2016a). This study also revealed that reductions in hippocampal power during rapid cycling VNS reached a saturation at approximately 500 μ A, suggesting that higher current outputs are not needed to produce maximal electrophysiological effects. Phase-amplitude coupling (PAC) is a measure of synchronization of different local neuronal populations in the brain and has important implications in epilepsy (Edakawa et al., 2016). It has been shown that PAC between theta and gamma frequency rhythms was markedly reduced during rapid cycle VNS at 250 μ A (Edakawa et al., 2016). Edakawa et al. (2016) showed that PAC during the ictal period (time during a seizure) was stronger than during the time between seizures (interictal period) in

drug-refractory epilepsy patients. Thus, the ictal suppression of PAC may be responsible for part of the anti-seizure effects of VNS and may be an important biomarker of improved seizure control. Rapid cycling VNS has been shown to reduce PAC in the healthy rodent brains (Larsen et al., 2016a). Hence rapid cycling may represent a specific way to maximize the benefit of PAC suppression in humans, and at a lower than normal output current; and at the same time improving efficacy, tolerability, and battery life. Preclinical studies have also laid the foundations for our understanding of the brain regions involved in the anti-epileptic effects of VNS. Early animal work by Naritoku et al. (1995), quantified the expression of fos, a protein marker for elevated neuronal activity, in rats to determine which brain regions are implicated in the anti-convulsant effects of VNS. Animals exposed to VNS (30 Hz, 500 μ s, 30 s on, 5 min off), showed activation of several limbic structures, including the amygdala, the diencephalon (including the habenula and lateral posterior thalamic nuclei), ventromedial and arcuate hypothalamic nuclei and in the brainstem, including the LC. Human positron emission tomography (PET) studies have also shown activation of these regions during VNS at clinically relevant parameters (Henry et al., 1998). Henry et al. (1998) demonstrated in epilepsy patients that acute VNS of the left cervical vagus nerve results in increases in regional blood flow in regions that are known to receive vagal input and generate vagal output, such as the rostral medullary region (which contains the NTS and DMNV), hypothalamus, thalamus, anterior insula, orbitofrontal cortex, inferior frontal gyrus and inferior parietal pole, with decreases in blood flow in the hippocampus, amygdala and posterior cingulate gyrus. In humans, functional imaging studies have shown a clinical heterogeneity in the brain regions which appear to be activated or suppressed during VNS (Chae et al., 2003). This may be due to small sample sizes, varying epilepsy types and aetiologies, differing duration of disease, varying duration of treatment with VNS, and varying anti-seizure medications. Perhaps the strongest factor impacting the clinical variance in the activation and suppression of higher order brain regions is the time since implantation and activation of VNS devices at which these brain imaging studies have taken place (Chae et al., 2003). Functional imaging performed on individuals who have had chronic VNS therapy (months to years) show altered brain functionality when compared with patients who are imaged almost immediately post-initiation of VNS. The implication here is that after a period of time, VNS results in changes in activity within different brain regions. Further imaging investigations in a clinical setting are required at different time points post-VNS implantation to track any changes in neuronal activity that may result with chronic stimulation therapy.”

I also think the manuscript could be improved by more discussion over why there are decreases in SUDEP with VNS - what can we learn from this? Are there less deaths due to arrhythmia for example? Linked to this I think it would also be beneficial to talk more about how the VNS impacts control of the heart and sympathetic innervation. With this in mind, does slowing of AV conduction with VNS increase the risk of long QT intervals?

We thank the reviewer for their comment. We do not fully understand the mechanism behind why there are less deaths of SUDEP with VNS. One speculation is that this may be due to an alteration in the pattern of autonomic activity, restoring balance between sympathetic and parasympathetic activation. Ongoing research is being conducted to try and determine the exact reasons for this phenomenon. We have included a discussion on HRV and how it has been shown to be a predictor for SUDEP and how VNS may be implicated in reducing the risk of SUDEP (lines 357-377).

“...Proposed mechanisms behind SUDEP include hypoventilation or a cardiac dysrhythmia provoked by the onset of a seizure, though it has been suggested that SUDEP can occur from different mechanisms in different people as reports of sudden death unrelated to seizures in epilepsy patients have been made (Shorvon & Tomson, 2011). Heart rate variability (HRV) is routinely used as an indirect measure of autonomic control over the cardiac cycle (Malliani et al., 1991). While altered HRV has been shown to be a predictor for sudden death in certain diseased states, such as heart failure (Kleiger et al., 1987), its association with SUDEP as a risk factor is unclear. However, it is thought that autonomic dysfunction or altered HRV contributes to an increased risk of SUDEP (Myers et al., 2018). In an attempt to characterize the association of altered HRV as a risk factor of SUDEP, Sivathamboo et al. (2021) conducted a multicentre, retrospective, case-control study to compare HRV in SUDEP cases and living epilepsy patients. Sivathamboo et al. (2021) used ECG to calculate HRV and reported time and frequency domain components, the latter including low-frequency power (LFP) (0.04-0.14Hz) and high frequency power (HFP) (0.15-0.40Hz). HFP is thought to represent vagal cardiac tone whereas, LFP is considered a reflection of sympathetic and vagal cardiac tone (Shaffer & Ginsberg, 2017). The results from Sivathamboo et al. (2021) showed that increased HFP was associated with longer survival in SUDEP. As HFP is thought to represent vagal outflow to the heart, the finding by Sivathamboo et al. (2021) supports the cardioprotective effects of increased cardiovagal tone. Importantly, this may contribute to the decreased rates of SUDEP seen in drug-refractory patients treated with vagus nerve stimulation (Ryvlin et al., 2018).”

Line 406 - if the left vagus nerve is targeted for VNS - then why would the authors anticipate it has an effect on HRV - when the SA node seems to be responsible for driving the variability? Perhaps some comment on this somewhere would be helpful. Interestingly - atropine does not have an affect on AV conduction. <https://pubmed.ncbi.nlm.nih.gov/7733345/>

We thank the reviewer for their comment. We don't anticipate an effect on HRV as the left vagus nerve is stimulated and as we have previously stated *“When comparing right and left VNS in the rabbit, it was shown that right VNS was more effective at producing bradycardia than left VNS (Brack et al., 2004).”*(lines 199-201).

Minor comments:

Intro - spleen important organ innervated by the vagus - potentially add this?

We thank the reviewer for their comment. We note that direct vagal innervation of the spleen is debated though there is evidence of indirect innervation, thus we have not included it as one of the organs directly innervated by parasympathetic vagal fibers in this review (Bassi et al., 2020).

Line about GVA (108) innervating aortic chemoreceptors - isn't this also true for the carotid chemoreceptors and baroreceptors?

We thank the reviewer for their comment. The carotid sinus nerve, a branch of the glossopharyngeal nerve carries carotid chemoreceptor and baroreceptor afferents (Wang *et al.*, 2019), whereas GVA within the vagus innervate the aortic chemoreceptors and baroreceptors.

Line 135 - define NA and DMNV

We thank the reviewer for their comment. These terms are defined in the preceding section (lines 114-115):

“...the nucleus ambiguus (NA)... dorsal motor nucleus of the vagus (DMNV)...”

Line 131 - afferent projections of the vagus - shouldn't this section be vagal afferent network within the brain (otherwise people might expect this section to be about afferent innervation of other organs)

We thank the reviewer for their comment. We have amended the title as follows (line 130):

“Vagal afferent network within the brain”

Line 198 - Vagus also promotes lusitropy? (rate of relaxation)

We thank the reviewer for their comment. We have now included this point as follows (lines 219-220):

*“Stimulation of the vagus nerve has also been shown to promote lusitropy (Henning *et al.*, 1996).”*

Line 237 - also the carotid chemoreceptors

We thank the reviewer for their comment. We have included this as follows (lines 261-262):

“...and that carotid chemoreceptor activity is elevated in the HF state...”

REQUIRED ITEMS

- Please include an Abstract Figure file, as well as the Figure Legend text within the main article file. The Abstract Figure is a piece of artwork designed to give readers an immediate understanding of the Review Article and should summarise the main conclusions. If possible, the image should be easily 'readable' from left to right or top to bottom. It should show the physiological relevance of the Review so readers can assess the importance and content of the article. Abstract Figures should not merely recapitulate other figures in the Review. Please try to keep the diagram as simple as possible and without superfluous information that may distract from the main conclusion of the Review. Abstract Figures must be provided by authors no later than the revised manuscript stage and should be uploaded as a separate file during online submission labelled as File Type 'Abstract Figure'. Please ensure that you include the figure legend in the main article file. All Abstract Figures will be sent to a professional illustrator for redrawing and you may be asked to approve the redrawn figure before your paper is accepted.

- Please upload separate high quality figure files via the submission form.

- Author profile(s) must be uploaded via the submission form. Authors should submit a short biography (no more than 100 words for one author or 150 words in total for two authors) and a portrait photograph of the two leading authors on the paper. These should be uploaded and clearly labelled together in a Word document with the revised version of the manuscript. Any standard image format for the photograph is acceptable, but the resolution should be at least 300 DPI and preferably more. A group photograph of all authors is also acceptable, providing the biography for the whole group does not exceed 150 words.

- It is the authors' responsibility to obtain any necessary permissions to reproduce previously published material and to list these within the main article file. For information, please see: https://jp.msubmit.net/cgi-bin/main.plex?form_type=display_requirements#permissions.

END OF COMMENTS

- Bassi GS, Kanashiro A, Coimbra NC, Terrando N, Maixner W & Ulloa L. (2020). Anatomical and clinical implications of vagal modulation of the spleen. *Neurosci Biobehav Rev* **112**, 363-373.
- Brack KE, Coote JH & Ng GA. (2004). Interaction between direct sympathetic and vagus nerve stimulation on heart rate in the isolated rabbit heart. *Exp Physiol* **89**, 128-139.
- Wang FB, Liao YH, Kao CK & Fang CL. (2019). Vagal baro- and chemoreceptors in middle internal carotid artery and carotid body in rat. *J Anat* **235**, 953-961.

Dear Vaughan,

Re: JP-TR-2024-287164R1 "The physiology, anatomy and stimulation of the vagus nerve in epilepsy" by Mikaela Patros, Shobi Sivathamboo, Hugh David Simpson, Terence J O'Brien, and Vaughan G Macefield

We are pleased to tell you that your paper has been accepted for publication in The Journal of Physiology.

Please see below for a couple of comments that can be addressed at proof stage.

Authors should note that it is too late at this point to offer corrections prior to proofing. Major corrections at proof stage, such as changes to figures, will be referred to the Editors for approval before they can be incorporated. Only minor changes, such as to style and consistency, should be made at proof stage. Changes that need to be made after proof stage will usually require a formal correction notice.

Best wishes,

Laura Bennet
Senior Editor
The Journal of Physiology

P.S. - You can help your research get the attention it deserves! Check out Wiley's free Promotion Guide for best-practice recommendations for promoting your work at www.wileyauthors.com/eeo/guide. You can learn more about Wiley Editing Services which offers professional video, design, and writing services to create shareable video abstracts, infographics, conference posters, lay summaries, and research news stories for your research at www.wileyauthors.com/eeo/promotion.

IMPORTANT NOTICE ABOUT OPEN ACCESS: To assist authors whose funding agencies mandate public access to published research findings sooner than 12 months after publication, The Journal of Physiology allows authors to pay an Open Access (OA) fee to have their papers made freely available immediately on publication.

You can check if your funder or institution has a Wiley Open Access Account here: <https://authorservices.wiley.com/author-resources/Journal-Authors/licensing-and-open-access/open-access/author-compliance-tool.html>.

EDITOR COMMENTS

Reviewing Editor:

Thank you for your revisions which were well received. Note there is one minor suggestion on terminology from reviewer 2. Additionally, please remove the reference in the abstract. Both of these can be done at page proof stage.

Senior Editor:

Thank you for a wonderful review.

REFEREE COMMENTS

Referee #1:

Thank you for taking the time to respond to all of my comments on the initial manuscript. I do not have any other comments. I hope this review is well read and cited widely.

Referee #2:

Thank you for answering all my comments.

I only have one additional comment about the added content - in the SUDEP section, the authors indicate that SUDEP is caused by "hypoventilation". But this terminology isn't correct, respiratory arrest doesn't happen because of hypoventilation, it happens because of apnoea. Perhaps respiratory arrest and apnoea might be a better term to use?